# Drugs That Changed Society: History and Current Status of the Early Antibiotics: Salvarsan, Sulfonamides, and β-Lactams

**DOI:** 10.3390/molecules26196057

**Published:** 2021-10-07

**Authors:** Søren Brøgger Christensen

**Affiliations:** The Museum of Natural Medicine & The Pharmacognostic Collection, University of Copenhagen, Jagtvej 162, DK-2100 Copenhagen, Denmark; soren.christensen@sund.ku.dk; Tel.: +45-3533-6253

**Keywords:** bacterial infectious diseases, salvarsan, sulfonamides, penicillins, cephalosporins, carbapenems, thiopenems, monobactams, β-lactamases

## Abstract

The appearance of antibiotic drugs revolutionized the possibilities for treatment of diseases with high mortality such as pneumonia, sepsis, plaque, diphtheria, tetanus, typhoid fever, and tuberculosis. Today fewer than 1% of mortalities in high income countries are caused by diseases caused by bacteria. However, it should be recalled that the antibiotics were introduced in parallel with sanitation including sewerage, piped drinking water, high standard of living and improved understanding of the connection between food and health. Development of salvarsan, sulfonamides, and β-lactams into efficient drugs is described. The effects on life expectancy and life quality of these new drugs are indicated.

## 1. Introduction

Infectious or communicative diseases have been a severe burden for mankind for millennia. The absence of knowledge of the pathogen organisms prevented prophylaxis and treatment of the maladies until the early 21st century. In Denmark, an overview of morbidity reveals that in the period 1876–1880 communicative diseases were the cause of almost 3/4 (72%) of mortality, in the period 1900–1903 almost half the deaths (42%), in 1920 about a 1/7 (15%), in 1945 only 4% and in 2013 only 0.5% [1,2]. The abrupt fall in death rate of communicative infections was caused by sanitation, higher living standards and introduction of efficient drugs as will be explained below. Bacteria were described as animalcules already in 1683 by the Dutch merchant Leeuwenhoeck [3,4]; but the importance of the discovery was not realized until 200 years later. Several different microorganisms were described and depictured by a Danish scientist Müller in the atlas *Animalcula infusoria fluviatilia et marina* (1786) [5]. The first correlation between microorganisms and health came in 1836 when the Italian civil servant Bassi described that a minute fungus (*Botrytis bassina*) was fatal to silkworms. The German clinician Shönlein realized in 1839 that the pathogen causing the human skin disease favus was a fungus (*Achorion schöleinii*). Some scientists proposed in the 1830s that fermentation was a conversion of organic compound performed by living microorganisms (yeast). However, leading chemists such as Berzelius and Liebig suggested that fermentation was a purely chemical process [3]. In 1873, Pasteur published a study on the diseases of the vine revealing that in the microscope he could see two organisms on the surface of the peel of grapes, some small organisms and some big organisms. By heating the grapes to 80 ℃ the small organisms were killed but the big survived. This later has been called Pasteurization [6]. Today the small organisms are known as bacteria and the big organisms as yeast. Another important contribution of Pasteur was the demonstration that life does not evolve spontaneously. Development of life in uninfected media was demonstrated to originate from germs present in the air. In 1849, Davaine demonstrated the presence of *Bacillus anthracis* in the blood of cattle suffering from anthrax. This was the first mammalian disease for which a bacterium was shown to be the pathogenic organism. Pasteur showed in 1877 that transfer of blood from an infected animal to a healthy animal communicates the disease. Pasteur also realized that virulence of the anthrax bacteria could be attenuated by cultivation at high temperature. Inoculation in animals with the attenuated organism protected the animal from developing disease. The technique of decreased virulence was used for prevention of rabies in a boy bitten by a mad dog. In 1882 Koch discovered the tubercle bacterium (*Mycobacterium tuberculosis*). The studies of Pasteur and Koch formed the foundation of bacteriology as we know it today [3]. After these pioneering studies several pathogenic bacteria were characterized, *Corynebacterium diphteriae* (diphtheria) *Pasteurella pestris* (plague), *Clostridium tetani* (tetanus), *Salmonella typhosa* (typhoid fever), and *Mycobacterium tuberculosis* (tuberculosis). In 1900, pathogenic organisms causing 21 diseases were discovered [3]. A late discovery of new pathogenic agents was made in 1983 when bacteria were discovered in the stomach of patients suffering from gastritis and peptic ulceration [7,8]. The bacteria later were identified as *Campylobacter pyloridis* [9]. The discovery was met with some skepticism, since it was difficult to accept that bacteria could live in the hostile environment of the stomach [10]. In 2008, Warren and Marshall were awarded the Nobel Price for their discovery of the pathogenic properties of what today is known as *Helicobacter pylori* [10]. In 2001, several 1425 infectious organisms were known, including 217 virus and prions, 538 bacteria, 307 fungi, 66 protozoa, and 287 helminths [11].

The importance of hygiene for life expectancy was gradually understood in the last part of the 19th century. The ideal man in Greek/Roman philosophy was healthy and beautiful. To obtain a beautiful body, public baths were needed. Aquaducts provided fresh and clean water and public latrines were used. Sewers (cloacae) build in the 6th century B.C. are still a part of the Roman sewerage. According to Christianity, the body is just the holster for the soul and consequently needed no attention [12]. After the fall of the Roman Empire in the 5th century A.D., hygiene in Europe declined. In contrast, it was to some extent maintained in Medieval Muslim Europe [13]. Urbanization as a consequence of industrialization afforded metropoles with poor hygiene. Two major cholera outbreaks (1848–1849 and 1865–1867) encouraged sanitation. Construction of sewerages in most major European cities were completed in the last decade of the 19th century [13]. It soon was realized that dumping of the superfluous water and filth in rivers or seas caused unacceptable contamination. Consequently biological filters were installed in the early 21st century [13]. Sanitation was a time consuming project, even in the 1970s apartments with shared latrines in the backyard with no drainage could still be found in Copenhagen and probably also in other European cities. It is encouraging to realize that the construction of sewerage was initiated after the big cholera epidemics, but before it was established that the cholera bacteria (*Vibrio cholerae*) was the pathogenic organism 1883 [3]. Supply of clean piped water in general developed slower since this very often was constructed by private enterprises [13,14]. In northern Europe most cities had piped water in 1915 and most cities in southern Europe in 1940. Later the importance of higher standards of living and proper food has been realized [3].

The effect of sanitation is clearly revealed by the abrupt fall in mortality caused by infectious diseases in the period 1875 to 1920. However, one infectious disease remained unaffected: syphilis [1]. Efficient therapy for bacterial diseases slowly started to appear in the second decade of the 20th century but the breakthrough came when sustainable production of penicillin was invented in the 1940s. The naturally occurring β-lactams afforded access to efficient antibiotic drugs [15,16,17]. The term chemotherapy was introduced by Ehrlich. It was defined as the use of drugs to injure an invading organism without injure to the host [18]. Later antibiotics were defined as compounds combating invading organisms [19]. Today the term chemotherapeutics is mainly used for drugs curing cancer diseases, in contrast to the original definition of Ehrlich. Antibiotics are drugs curing infectious diseases, irrespectively of natural or synthetic origin. The years from about 1940 to 1960 were the golden age of antibiotics. In this period most of the antibiotic classes used today were introduced. Emerging resistance toward antibiotics, however, becomes an increasing threat. In 2012, the Congress of the USA passed legislation for antibiotic development. In 2014, president Obama issued an order for substantial government intervention to stimulate infectious disease research and drug development [20]. Not all bacteria are hostile. An intact intestinal microbiota affords and important protection against pathogenic bacteria. Antibiotics should be used with care to prevent corruption of the intestinal flora [21]. The difference between antibiotics and vaccination (Section 9) is that the former only cure a disease, whereas the latter primarily prevents infection.

## 2. Arsphenamine (Salvarsan)

The number of deaths of communicative diseases dramatically decreased in the period from 1875 to 1903. In contrast, the morbidity of syphilis remained approximately 0.2% of all deaths in Denmark [1]. Syphilis, “the terror of the early 20th century [22]”, was a heavy burden in other parts of Europe, with 1/6 of all Parisians and 1/10 of all Londoners infected [22]. The first efficient drug was based on a wrong but fruitful working hypothesis made by Paul Ehrlich: “Some dyes specifically color some organs and microorganisms. Consequently, these dyes can be used for selective targeting these organs or organisms.” Based on this observation Ehrlich concluded that a magic bullet (*die magische Kugel*) can be designed. The magic bullet enables selective killing of microorganisms or cancer cells. Ehrlich tested several compounds for their ability to cure syphilis infected mice. In 1909, compound 606, arsphenamine or salvarsan (Figure 1) turned out to have an effect. Already a year later positive clinical results were published [23]. The synthetic path for preparation of arsphenamine occasionally resulted in toxic contaminants [24]. Thus, after only a few decades the drug was displaced with the safer penicillin [22]. In most databases and textbooks arsphenamine is depicted as **1a** (Figure 1). However, ESI MS has revealed that the molecule is a mixture of cyclic compounds **1b**, with different numbers of arsenic atoms in the ring. The dominating molecules are trimers of 4-hydroxy-3-aminophenylarsen (**1b**, n = 1) and pentamers (n = 3), but also tetramers, hexamers and heptamers (n equals 2, 4 and 5) are present [24]. Arsphenamine was marketed as salvarsan. Poor aqueous solubility of arsphenamin was overcome by preparation of neosalvarsan, for which the structure **2a** has been suggested [18]. Later studies suggested that the cyclic arsenicals only are prodrugs of the active warhead osphenarsine **3** [24]. Arsphenamine was the first marketed antibiotic, which cured an infectious disease caused by bacteria [25]. For a period until the 1940s, oxphenarsine (**3**) replaced salvarsan under the tradename Mapharsen until only penicillin was used for treatment of syphilis [18].

Arspehnamine was not the first example of a magic bullet, quinine used for the treatment of malaria had been known for almost a century [26]. 

## 3. Sulfonamides

### 3.1. Discovery

Ehrlich’s idea of a magic bullet inspired G. Domagk to test several hundred compounds. Azo compounds gave promising results when tested on mice infected with streptococci, but were inactive in vitro [27]. On December 20, 1932 prontosil (Figure 2, **4**) was shown to prevent mice and rabbits from dying of a lethal infection of *Streptococcus* [28]. Soon after filing of the patent clinical studies were performed. Patients suffering from infections with *Streptococcus* were treated. Encouraging positive results were obtained in most cases, although severe infections did not respond satisfactory [29,30]. Domagk’s daughter at six years of age developed a life threatening infection, which in a miraculous way was cured by administration of prontosil [22]. The compound developed into a blockbuster for I. G. Farben, at which Domagk was employed. In 1935 the sale of Prontosil totaled 175,000 marks, 1936 the sale rose to about 1 million marks and in 1937 more than 5 million marks [27]. Soon the drug was introduced to the world. It was considered a “miracle drug”. Since German patents were not recognized in France, an independent production was made in this country under the name of rubiazol. A side effect was that the skin of some patients became reddish. Fortunately this effect was temporary. The introduction of the sulfa drugs was a turning point in treatment of streptococcal diseases such as childbed fever and septicemia, pneumonia, meningitis, dysentery, gonorrhea, and urinary tract infections [27]. Already in 1939, Domagk was offered the Nobel Price, but the situation in Germany prevented him from receiving it until 1947 [22].

By accident Bovet’s group in 1935 tested 4-aminobenzenesulfamide (sulfanilamide, **5**) for treatment of streptococcal infections in rabbits. To their surprise and disappointment **5** was as efficient as prontosil. However, some reflections led to the conclusion that the discovery of a simple but efficient molecule opened the possibilities for designing new simple antibiotics [31]. Shortly thereafter a new study revealed that in contrast to prontosil 4-aminobezenesulfonamide inhibits the growth of bacteria in vitro and is efficient for treatment of streptococcal infections in man [32]. It is now accepted that prontosil (**4**) is a prodrug for 4-aminobenzenesulfonamide (**5**) formed by reductive cleavage of the N=N double bond [33]. 

In 1938, May and Baker marketed 4-sulfapyridine (**6**) as the first sulfa drug, which could cure “captain of the death“ bacterial pneumonia. In late 1943 Winston Churchill suffered from a potentially lethal pneumonia. The infection was miraculously cured by 4-sulfapyridine (**6**), a sale promoting event [27]. The introduction of sulfa drugs for treatment of infections revolutionized treatment of streptococcal infections in the early 20th century [34]. The introduction of penicillin decreased the demand for sulfa drugs [27]. 

### 3.2. Mechanism of Action

If bacteria are incubated with yeast extract containing 4-aminobenzoate and sulfanilamide (**5**) the growth inhibition is antagonized (Figure 1, **8**) [35]. This discovery led to understanding of the mechanism of action of sulfanilamide. Tetrahydrofolate (**10**) is formed from the building blocks 7,8-dihydropteridinepyrophosphate (**7**) and 4-aminobenzoic acid (**8**) in bacteria [36]. Sulfonamide (Figure 3, **5**) interacts with dihydropteroate synthase and prevents formation of tetrahydrofolate (**10**). If sufficient 4-aminobenzoate is present sulfonamide will be displayed from the enzyme neutralizing the inhibition. Tetrahydrofolate (**10**), a precursor for folic acid, is a vitamin for mammalian [37]. Consequently, administration of an antimetabolite of 4-aminobenzoic acid (**8**) does not affect mammalian cells. If no other targets for sulfanilamide than the tetrahydrofolate existed, the compound would be harmless for mammalians, whereas it is a potent proliferation toxin for microorganisms. 

The sulfonamides have two proteolytic groups the sulfonamide with a pK_A_ value of 5–8 and the protonated amino group pK_A_ 2–2.5 [36]. Poor aqueous solubility complicates administration of sulfa drugs. Consequently, they are administered as the more water soluble sodium salts.

Side effects of sulfonamides or accidentals discoveries have led to several drugs possessing the pharmacophore, but with no antibiotic effects. Examples of such drugs are acetazolamide and furosemide (both used as diuretics), glibenclamide (for treatment of diabetes mellitus 2), parecoxib (a COX2 inhibitor) and dofetilide (a class III antiarrhythmic drug) [36]. 

## 4. β-Lactams

Six ring systems including a β-lactam ring are found in nature β-lactams, penams, cephems, clavan, penems or thiopenem and carbapenems (Figure 4) [37]. The clinical use of β-lactams has revolutionized treatment of infectious diseases [38]. The systematic name for the β-lactam ring is azetidinone.

### 4.1. Discovery of the Penicillins

If untreated infections with *Streptococcus aureus* might be fatal. In the first decades of the 20th century, no drug for treatment was known [39]. By accident, studies of the rare mold *Penicillium rubrum* were performed in laboratories in the vicinity of Flemings laboratory. Some spores appeared on a Petri dish used for growth of *Staphylocccus sp.* This dish was left by Fleming in his laboratory during a holiday. During his absence an unusually cold period in London favored the growth of *p. rubrum*. When the mold culture was well established, the temperature rose favoring growth of *S. aureus.* Inhibition zones around the mold colonies led Fleming to the conclusion that antibacterial compounds were produced, a simple but far reaching conclusion [22]. The growth of several other bacteria such as *Staphylococcus pyrogenes*, *Streptococcus pyrogenes*, *Streotococcus viridans*, diphteroid bacillus, and *Micrococcus lysodeikticus* was also inhibited, but not the growth of *Bacillus anthracis*, *Bacillus typhoeus*, and enterococcus [40]. In his publication from 1929, Fleming claimed that *p. rubrum* had contaminated his dishes [40]. Later publications, however, mention *p. notatum* [22,41]. Fleming used the broth of *p. rubrum* to cure a severe conjunctivitis in a student [22]. These data, surprisingly, did not call for attention. This situation changed when Florey at the University of Oxford took up the challenge to develop a protocol for large scale production of penicillin. Florey supervised a group consisting of two biochemists Chain and Heatley. Industrial large scale production of penicillin, however, first succeeded when Florey and Chain went to the USA. They constructed 500 gallon tanks, replaced the growth medium with corn steep liquor and found a new *Penicillium* species (*p. chrysogenum*) producing 100 times more penicillin. These changes enabled production of penicillin in ton scale. By up scaling the equipment to 7500 gallon tanks Pfizer produced 4 tons in 1945 [22,41,42]. The results from the first clinical trials were so impressive that the US army immediately included penicillin in the armamentarium for combating infections in the military hospitals [43]. The US army, being the main sponsor, received all the produced penicillin until the production in 1945 fulfilled the army’s demand. The surplus enabled civilians to benefit from the drug. F-pencicllin (Figure 5, **22**) and I-dihydropenicilllin (**23**) were the first isolated natural products possessing the penam nucleus. The *N*-acyl side chain originates from carboxylic acids in the medium. The phenylacetyl group in G-penicllin (**24**) was introduced after shifting the medium to corn steep liquor [44]. Addition of different carboxylic acids to the medium enables variation of the *N*-acyl group [42,44]. Even today after development of sophisticated syntheses for azetidinone the majority of pencillins marketed are made by semisyntheses using starting materials obtained from *Penicillium* broths [44,45].

In addition to the impressive research performed in UK and USA a small pharmaceutical company in Denmark, Løvens Kemiske Fabrik (today Leo Pharma), succeeded in production of penicillin already in 1943. The prepared penicillin was used in some clinical cases. The late Danish Minister of Foreign Affairs, Per Stig Møller, was cured for pneumonia. Production was shut down in 1944 to prevent the Nazi occupation forces from accessing the strategic valuable drug [46]. After the war, the production was reestablished and the company became an important supplier to the market. 

### 4.2. Structure Elucidation and Chemistry of Penicillin

The structure of penicillin was established through several chemical reactions (Figure 2 and Figure 3) and by X-ray crystallography. The structure was published in two almost identical manuscripts in Nature and Science authored by research groups from several Anglo-American pharmaceutical companies. From a historical point of view the two publications are instructive examples on structure elucidation before the development of spectroscopy [47,48]. In the 1940s, the challenge of elucidating the structure of a crystalline compound with molecular weight of about 350 required enormous intellectual and financial resources as evidenced by the list of contributors to the two manuscripts [47,48,49]. Hodgkin published an X-ray analysis of a crystalline sodium salt of **22** confirming the previously reported results [50]. Several misleading structures were suggested before the correct structure was established [49]. Today, an average master’s student might elucidate the structure in weeks, taking advantage of developed spectroscopy tools. The 2D NMR pulse sequences developed by the recently deceased Nobel laureate Richard R. Ernst in particular has facilitated structure elucidation [49].

Tension in the β-lactam ring causes the lactam to be extremely reactive compared to other amides. The stability of amides is partly explained by the two resonance structures the amide and the zwitterionic structure. Tension in the four-membered lactam ring makes the zwitterionic resonance structure unfavorable (Figure 4) [45,51]. As a consequence, penicillin reacts freely with alcohols. In penicillin-binding peptides a serine residue in the active site is esterified by the penicillins [52]. Other examples of reactive cyclic amides are known (see Section 9) [53].

From a biogenetic point of view penicillin is a non-ribosomal peptide originating in the peptide ζ-L-aminoadipidoyl-L-cysteinoyl-D-isovalerine, which by the enzymes isopenicillin *N*-synthase and isopenicillin N *N*-acyltransferase is converted into penicillin G (Figure 5, **24**) [37,54]

A major step in developing later generations of penicillins was discovery of 6-aminopenicillanic acid in a broth used for production of the penicillins **22**–**26** (Figure 6, **38**). Despite the level of molecules possessing the β-lactam ring the broth revealed an unexpected low antibiotic effect. This discrepancy was solved by the poor antibiotic effect of isolated **38** [55]. Access to the free amine facilitated syntheses of several penicillin analogues enabling development of later generations of penicillins [38,42,56,57].

### 4.3. Mechanism of Action of Penicillins

The cell wall or cell envelope is and essential polymer surrounding and enclosing bacteria, Defects in the polymer might be lethal [52]. Several enzymes and transporters are involved in the construction of the cell wall including several penicillin-binding peptides (PBP). The cell wall of *Staphylococcus aureus* is fortified by crosslinking strands of peptidoglycans with strands of peptides as illustrated by the cartoon shown in Figure 7 [52,58]. The L-lysine residue is replaced with meso-diaminopimelic acid in other bacteria [44]. 

The cross binding occurs by elimination of the terminal D-alanine in one peptidoglycan strand by an attack of the amino bond of a glycine residue terminating a side chain in another peptidoglycan strand (Figure 8) [52]. Hereby a cross linking of the two peptidoglycan strands is obtained. This mechanism deviates from pervious described mechanism for cross binding [44]. 

The penicillin nucleus is isoster to the terminal D-Ala-D-Ala unit essential in the cross binding process (the Tipper-Strominger hypothesis) [59]. By interacting with and acylating the serine hydroxyl group in the penicillin-binding transpeptidases [60] cross binding and consequently formation of a stable cell wall in the bacteria is prevented [44,52]. It was hypothesized that the 6α-methyl analog of penicillin was a closer analog to the D-Ala-D-Ala target. Surprisingly, the compound showed much poorer activity [61].

Gram-positive and Gram-negative bacteria differ in Gram-staining. Gram-negative bacteria possess an outer cell membrane, which prevents some antibiotics from reaching their target molecules. The outer membrane possess protein channels (porins) allowing small molecules such as nutrients or antibiotics to penetrate down a concentration gradient to reach the inner parts of the bacteria [38].

### 4.4. Late Generations Penicillins

The first isolated penicillins were characterized by simple acyl groups at *N*-6 (Figure 5). All of these have the drawback that they to a large extent are degraded in the acidic medium in the stomach. In addition they only show activity against a narrow spectrum of bacteria. Access to 6-aminopenicillanic acid (**38**) facilitated synthesis of analogues with different acyl groups. Today **38** is prepared by enzymatic cleavage of penicillin G (**24**) [44,55]. Compound **38** had previous been mentioned but the potential was never realized by the Japanese scientists [62]. Early successes with semisynthetic penicillins were obtained by preparing amides of phenoxyacetic acids (**39**–**41**) [55]. The compounds are also named phenoxyalkyl penicillins. The compounds **40** and **41** are administered as racemic mixtures. The phenoxyacetamides are more acid resistant and are adsorbed about four times more efficient from the stomach [63,64]. The presence of an oxygen atom β to the carbonyl group makes the side chain carbonyl much less electrophilic and prevented reaction with the lactam carbonyl in acidic media [38,42]. The compounds were developed in the late 1950s [42,55]. Phenoxymethyl penicillin (**39**–**41**) and penicillin G (**24**) are used orally for treatment of strains of Gram-positive cocci [65]. The compounds with simple carboxylic acids in the side chain are classified as first generation penicillins [37,66]

The second generation penicillins consists of aminopenicillins such as **42**–**45** (Figure 9). Introduction of the amino group, which will be protonated in acidic media, makes the compounds less acid labile [38]. These penicilllins are active against Gram-positive as well as Gram-negative bacteria such as *Haemolhillus influenza, E. coli* and *Proteus mirabilis*. They have good bioavailability after oral administration [20,43,44]. Ampicillin was the first penicillin with some activity against Gram-negative bacteria [55]. The introduction of a hydroxyl group in the para position of the phenylglycine residue of ampenicillin to give amoxicillin (**44**) improved the absorption after oral administration. Amoxicillin was marketed in 1964 [38] and became one of the most widely prescribed penicillins [67]. The aminopencillins were developed in the early 1960s [55].

Introduction of an acidic group into the side chain made the penicillins **46**–**48** (Figure 10) clinically active against *Pseudomonas*
*aeroginosa* and some *Klebsiella* infections [67] (third generation penicillins). Piperacillin (**49**) and mezlocillin (**50**) with bulky side chains are less sensitive toward β-lactamases and generate fourth generation penicillins [66] (Section 4.6). These penicillins also were efficient in treatment of *P. aeruginosa* infections in immunocompromised patients [67]. The drug is discontinued in the USA [43]. The ureido group in the side chain of piperacillin enhances cell wall penetration of Gram.negative bacteria affording activity also against species belonging to *Pesudomonas*, *Klebsiella*, *Enterobacter* and *Citrobacter* [20]. The drugs were developed in the late 1960s and the early 1970s [55].

### 4.5. Penicillin Hypersensitivity

Due to reaction with the ε-amino group of lysine residues in proteins or disulfide formation between cysteine residues and degradation products of pencillins some patients may develop hypersensitivity following administration of penicillins. Fortunately most allergic reactions are benign [38]. Polymerization of ampicillin molecules caused by reaction of the amino group and the β-lactam may also result in allergens [38].

### 4.6. Penicillin Resistance

Microorganisms may become resistant to antibiotics targeting intracellular pathways by developing mechanisms pumping drugs out of the cells [26]. This strategy is useless for Gram-positive bacteria for disarming penicillins, since their target is the crosslinking of the cell envelope outside the cell membrane [52]. However, some Gram-negative bacteria such as *Pseudomonas aeruginosa, E. coli and Neisseria gonorrhoeae* possess an outer membrane with multidrug effective efflux pumps [68]. This outer membrane also might prevent small molecule-like antibiotics to penetrate to the penicillin-binding proteins [43]. 

Already Fleming discovered that the growth of *Balantidium coli* and several other bacteria was resistant to penicillin G [40]. This observation was made before penicillin G (**24**) was taken to the clinic [69,70]. The resistant factor was transferred between different bacteria and was assumed to be an enzyme [71]. This assumption was confirmed by the discovery of the β-lactamases. These enzymes efficiently neutralize penicillins by opening the β-lactam ring to give penicilloic acid (**33**). Cleavage of the β-lactam ring removes the antibacterial effect [43,52,72]. In principle, this is the same acylation of a serine residue that occurs, when penicillins inactivates penicillin-binding proteins. However, the β-lactamases are not essential for the microorganism. Fast hydrolysis reactivates the enzyme enabling ring opening of other β-lactam rings [52]. Genes conferring resistance to antibiotics have been found in 30,000-year old organisms suggesting that resistance against antibiotics was invented by Nature millennia before mankind developed drugs against infectious diseases [70,73]. β-Lactamases are widely distributed [70,71,74,75] and have been studied intensive because of their ability to neutralize the clinical effect of penicillins. In several pathogenic organisms, expression of β-lactamases is provoked by β-lactam antibiotics [76]. The β-lactamases are divided into three classes A (penicillin-hydrolyzing), C (cephalosporin-hydrolyzing) and D (oxacillin-hydrolyzing) and Class B the metallo-β-lactamases [44,70,72,77]. Class C are primarily found in Gram-negative bacteria [44]. The cleaving catalyzed by serine-β-lactamases (Class A, C and D) involves a serine residue in the β-lactamases which react with the β-lactam to give an ester of the serine (Figure 4). The enzyme penicillin complex is hydrolyzed to penicilloic acid (**33**) and reactivated β-lactamase [44,72,77]. In the metallo-β-lactamases a hydroxide complexed with two zinc ions in the active site opens the β-lactam [72]. A detailed mechanism of action of the different β-lactamases is given in some reviews [58,72]. 

Two approaches were initiated to overcome resistance against penicillins (1) syntheses of penicillins that are resistant to β-lactamases (fourth generation penicillins) and (2) use of β-lactamase inhibitors (Section 8). Other factors affecting the effect of antibiotics are population of bacteria and biofilm formation. Bacteria in biofilms are protected by the presence of extracellular polysaccharides [78]. Old colonies of bacteria often have a slower rate of proliferation and consequently a less sensibility toward the antibiotics. In addition a higher population and diversity of bacteria increases the likeliness of β-lactamase producing species [43].

Methicillin (Figure 11, **51**), introduced to the clinic in 1960, was the first penicillin with reduced β-lactamase sensitivity [43,57,79]. Nafcillin (**52**) and oxacillin (**53**) with a steric bulk preventing hydrolysis followed later [38,44]. Evidence for the importance of the bulky methoxy groups is revealed by the sensitivity of homologous penicillins toward β-lactamases. Introduction of a methylene group between the dimethoxyphenyl and the carbonyl group in methicillin (**51**) increases the sensitivity of the molecules toward β-lactamases [38]. Methicillin (**51**) is not absorbed when given orally and cannot penetrate the outer membrane of Gram-negative bacteria8]. It is supplemented with nafcillin (**52**) and oxacillin (**53**) [38]. A drawback of isoxazolyl penicillins such as **53** is a high protein binding lowering the serum concentration [55,57]. Oxacillin (**53**) and nafcillin (**52**) are not approved in Japan, although they are standard therapy for treatment of infections with susceptible *S. aureus* and *S. epidermis* in the major part of the world [43]. Cefazolin is used instead [80].

Another mechanism used by bacteria for gaining resistance against penicillins takes advantage of mutating the penicillin-binding proteins into enzymes distinguishing the penicillins from the endogenic substrate the D-Ala-D-Ala residue [43,52,55,81]. An example of such an enzyme is PBP2a produced by *S. aureus* [43,55]. Expression of PBP2a is the major reason for resistance against β-lactams. A surprising observation is that despite expression of PBP2a a synergistic effect of clavanulate, a β-lactamase inhibitor (Section 8), and β-lactam antibiotics is observed for *S. aureus* [52]. 

Penicillins substituted at C-6 were obtained by hypochlorite oxidation of penicillin G to give a intermediate chlorine derivative, which in methanol is converted into the α-methoxy derivative (Figure 6, **54**) [82].

Development of the technique for 6-methoxylation enabled preparation of temocillin, developed in 1980 (Figure 12, **55**). Even though this compound has a long half live in the blood (about 5 h) the antibacterial activity is restricted to multiple resistant Gram-negative bacteria [44]. Formidacillin (**56**) is highly potent against Gram-negative bacteria including *Pseudomonas* sp. and has some activity against *Streptococcus* sp. but none against *Staphylococcus* sp3]. The compound, however, never was approved as a drug by the FDA. 

The ureidopenicillins azlocillin (Figure 13, **57**), mezlocillin (**58**) and piperazillin (**59**) were developed in the 1970s. They are efficient for treatment of infections with *Pseudomonas aeruginosa*, *Klebsiella* and some *Enterobacter* species. Unfortunately, they have a poor effect on infections of *Escherichia coli*, *Enterobacter* and *Salmonella* species [38,83].

Since the 1970s, very few new antibacterial drugs possessing the penam carbon skeleton have been introduced. A search in To Market To Market from 2019 to 2006 (Annual Reports in Medicinal Chemistry and Medicinal Chemistry Reviews) revealed no penicillins brought to the market. In contrast, several drugs in which penicillins are combined with β-lactamase inhibitors has been approved [58,84].

### 4.7. Penicillin Prodrugs

Prodrugs, in which the carboxylic acid has been masked, were used for improving absorption of ampicillin (**42**). The first prodrug was pivampicillin (Figure 14, **60**) marketed by Leo Pharma [57] in the 1970s [85]. Administration of pivampicillin afforded three time higher serum concentration of ampicillin and higher maximal serum concentration after one hour than after administration of an equimolar amount of ampicillin [86]. The pivalic acid released after cleavage of pivampicillin (**58**) to give ampicillin (**42**) might have an effect on the carnitine plasma level [87]. Additionally, talampicillin (**61**) [88], bacampicillin (**62**) [89] and lenampenicillin (**63**) [90] were absorbed efficiently.

## 5. Cephalosporins

### Discovery and Production of Cephalosporins

Cephalosporins contain the backbone cephem (Figure 4). Cephalosporin C was isolated from a wild strain of *Cephalosporium* in 1945 [91]. The molecular formula was published in 1955 [91,92] and the constitution **64** (Figure 15) in 1961 [93]. An X-ray analysis confirmed the constitution **64** [94]. Today *Acremomium chrysogenum* (formerly *Cephalosporium acremomium*) is primarily used for production of cephalosporins. Similar stereochemistry at the β-lactam ring in the penicillins and cephalosporins was established by a rearrangement of phenoxymethyl penicillin sulfoxide to a cephalosporin without changing the stereochemistry in the β-lactam ring (Figure 7) [95].

The naturally occurring cephalosporin **64** is converted to the free amine by reaction of the protected acid with phosphorous pentachloride followed by hydrolysis of the intermediate iminoether (Figure 8) [44,96,97]. Procedures for enzymatic cleavage of the amide avoiding hazardous reagents have also been developed [97].

From a biosynthetic point of view, cephalosporin-like penicillin is a non-ribosomal tripeptide originating from peptide ζ-D-aminoadipidoyl-L-cysteinoyl-D-isovalerine. However, after formation of the penicillin (Figure 5) the 2 position in D-2-aminoadipic residue is isomerized, a radical formation at one of the methyl groups in valine and a ring expansion occurs to give the cephalosporin nucleus. After oxidation and acetylation of the methyl group cephalosporin C (**61**) is formed (Figure 9) [37,54]. The penicillin-cephalosporin rearrangement described in Figure 7 shows resemblance to the biosynthesis of cephalosporin from penicillin N.

Like the pencillins the cephalosporins are classified in generations [66]. The first generation of cephalosporins (narrow spectrum) consists drugs such as cephalexin (Figure 16, **65**), cefadrine (**66**) and cefadroxil (**67**). First generation drugs have activity against Gram-positive bacteria such as *Staphylococcus aurea, Streptococcus* species, even though they are expressing β-lactamases, and a few Gram-negative bacteria e.g., *E. coli, H. influenza* and *Klebsiella* species. They are used against uncomplicated skin and soft tissue infections [37,98,99]. Drugs belonging to second generation (intermediate spectrum activity) are cefalclor (**68**) and cefuroxime (**69**). They possess some activity against aerobic Gram-negative bacteria such as species of *Moraxella*, *Neisseria*, *Salmonella*, and *Shigella* [98,99]. They are used against upper and lower respiratory tract infections, sinusitis, and otitis media [99]. 

Examples of third generation (broad spectrum activity) cephalosporins are cefotaxime (Figure 17, **70**), ceftazidime (**71**), ceftriaxone (**72**), and ceftizoxime (**73**) [38,98,99]. The introduction of a methoxime group (**74**–**76**) makes the compound substrate for a tripeptide transporter in the intestinal mucosoal membrane [38]. They have good activity against most Gram-negative bacteria and are less sensitive toward β-lactamase [99]. In addition, they possess some activity toward Gram-positive bacteria and some methicillin resistant strains [99]. Cefepime (**74**), cefpirome (**75**), and cefquinome (**76**) belong to the fourth generation of cephalosporins (broad spectrum activity). By introduction of an ammonium ion the ability to penetrate the outer membrane in Gram-negative bacteria is increased [38]. They are effective against several resistant bacteria such as Gram-positive cocci, *Streptococcus pneumoniae,* Entero-bacteriaceae, and *Pseudomonas aeruginosa* strains. They are used for treatment of meningitis [98,99]. Fifth generation cephalosporins (extended spectrum activity) encompasses ceftobiprole (**77**) and ceftraoline (**78**). Ceftobiprole (**77**) is a very broad spectrum cephalosporin with activity against Gram-positive cocci, including MRSA and methicillin resistant *Staphylococcus epidermidis* (MRSE), penicillin-resistant *Streptococcus pneumoniae*, *Enterococcus faecalis* and many Gram-negative bacilli including β-lactamases groups C producing *E. coli* and *Pseudomonas aeruginosa* [99]. Cefepime (**74**) has a γ-aminobutyric acid residue in the structure meaning that after penetrating the blood-brain-barrier it might have neurotoxic effects [38]. Ceftaroline fosamil (**78**) is an example of a fifth generation cephalosporin. It is used against multidrug-resistant *Staphylococcus aureus*, including MRSA, VRSA, and VISA [99]. Ceftaroline fosamil is a water soluble prodrug which in the organism is dephosphorylated to yield ceftaroline [100]. Ceftaroline fosamil was approved by the FDA (2010) and the European Medical Agency (2012).

Cephalosporin hypersensitivity is unusual. The seldom cases are in general benign. Cross reaction between penicillin allergy and cephalosporin allergy has not been observed [38].

## 6. Carbapenems

Carbapenems (Figure 4) are characterized by a backbone consisting of the same bicyclic ring system as penicillin, but possessing a double bond between C-2 and C-3 and a carbon atom in the 1 position. The two hydrogen atoms at C-5 and C-6 are trans disposed in clinical important carbapenems in contrast to the cis disposed C-5 and C-6 hydrogens in penicillins and the C-6 and C-7 cis disposed hydrogens in cephalosporins [101]. This might explain a lower sensitivity toward some β-lactamases [38,101]. The first isolated carbapenem was olivanic acid isolated from the broth of *Streptomyces clavuligerus* (Figure 18, **79**). Due to penetration problems this compound was not pursued [101]. The *R*-1-hydroxyethyl group at C-6 appears to be optimal for an antibiotic effect of carbapenems. As described in Section 3 and Section 4 the sidechain at C-6 in peniclllins and C-7 in cephalosporins might be substituted to improve the clinical effect [101]. Thienamycin (**80**) was isolated from the broth of *Streptomyces cattleya* [101]. The presence of an electrophilic β-lactam ring and a nucleophilic amino group makes thienamycin unsuitable as a drug because of intermolecular amid formation [38].

In general, the in vitro spectrum of activity of the carbapenems is broader than that of penicillins and cephalosporins. Imipenem (**81**), penipenem (**82**) and doripenem (**86**) are used against infections with Gram-positive bacteria. The introduction of the amidine group in imipenem (**81**) ensures protonation at physiologic pH preventing a cleavage of the β-lactam ring [38]. Imipenem was approved as a drug in 1985 [38]. Biapenem (**83**), meropenem (**84**), ertapenem (**85**) and doripenem (**86**) to some extend show activity against Gram-negative bacteria. Meropenem was approved in 1996 [38]. In combination, meropenem (**84**) and the β-lactamase inhibitor clavulanic acid (Section 3.1) have an effect on methicillin drug resistant *Mycobacterium tuberculosuis*, a bacterium otherwise resistant toward β-lactam antibiotics [101]. The presence of a β-methyl group at C-1 prevents the carbapenems from being hydrolyzed in the kidney by renal dihydropeptidase-1 [102] and consequently prolong the effect by reuptake [38,103]. Doripenem (**86**) and ertapenem (**85**) were approved in 2007 and 2001, respectively [38].

In contrast to other β-lactam antibiotics, large scale production of carbapenems is based on synthesis and not fermentation [104]. Pioneering efforts making the compounds available for clinical use were performed by the Merck group [105,106,107] supported by others [108]. Review of methods for constructing the azetidinone intermediates have been published [109].

Different strategies for developing resistance are (1) reducing the sensitivity of penicillin-binding enzymes for carbapenems by mutation, (2) altering porins, expression of efflux pumps or (3) expression of β-lactamases. In Gram-positive cocci resistance is typically caused by mutation of penicillin-binding proteins. In Gram-negative rods expression of β-lactamases, porin change or/and alterations of penicillin-binding proteins makes the bacteria more resistant [101]. 

In the clinic treatment of patients with community-acquired pneumonia with meropenem (**84**), ertapenem (**85**) or imipenem (**81**) combined with cilastatin, a dihydropeptidase-1 inhibitor has shown overall god response [103]. Patients suffering from nosocomial pneumonia responded more positive after treatment with meropenem (**84**) than after treatment with ceftazidime (**71**) combined with tobramycin [103]. Carbapenems also have shown good effects on hospital-acquired pneumonia and cystic fibrosis [103]. Complicated intra-abdominal infections have been treated successfully with imipenem (**81**), meropenem (**84**), ertapenem (**85**) and doripenem (**86**) [103]. Meropenem (**84**) and doripenem (**86**) as single therapy and iminipem (**81**) combined with cilastatin have been shown to have good effects on urinary tract infections [103]. Ertapenem (**85**), and meropenem (**84**) in combination with cilastatin have shown positive response in treatment of complicated skin and soft tissue infections [103]. The carbapenems are classified into three groups: Group 1 such as ertapenem (**85**) broad spectrum carbapenems with limited activity against non-fermentative Gram-negative bacilli but suitable for treatment of community-acquired infections. Group 2: imipenem (**81**), meropenem (**84**) and doripenem (**86**) are broad spectrum carbapenems, with activity against non-fermentative Gram-negative bacilli. They have less sensitivity to base promoted hydrolysis in solution. Group 3 carbapenems are clinical activity against resistant *S. aureus*. The latter group is still under development [110].

A drawback of the carbapenems is poor oral activity [110]. To overcome this problem a prodrug, tebipenem pivoxil (Figure 19, **88**) was prepared. The prodrug strategy is analogous to the principle used for pivampicillin (**59**) the prodrug of ampicillin (**42**). At the present this compound is under development as an orally active drug against pathogens causing respiratory tract infections [111]. Sanfetrinem (**89**) is the first tricyclic carbapenem. It possesses several interesting in vitro activities [110].

## 7. Thiopenems

The synthesis of faropenem (Figure 20, **90**) was published in a patents and Chinese journals between 2000 and 2010 [112]. The prodrug faropenem medoximil (**91**) is well absorbed after oral administration [38]. A synthesis of sulopenem (**92**) is published [113]. The key reaction is a triethoxyphosphine catalyzed thiopenem cyclization of the intermediate **94** (Figure 10). Sulopenem is also administered as a prodrug sulopenem etzadrozil (**93**) [38]. Introduction of a C-C double bond in the thiazolidine ring and the presence of a sulfur atom reduce ring tensions and reduce the reactivity of the β-lactam ring. The thiopenems are active against Gram-positive *cocci*, *Streptococcus pneumoniae* and Gram-negative bacteria but not against *Pseudomonas aeruginosa* [38,114]. No thiopenem are found on the homepage of the European Medical Agency (August 2021) indicating that no thiopenem is approved as a drug. The FDA previously have rejected the registration of faropenem [115].

## 8. Monobactams

The monobactam goup is the last discovered types of β-lactams. The first description is in a Japanese patent from 1979. For structure elucidation sulfazecin (Figure 11, **95**) isolated from the broth of *Pseudomonas* species was converted to D-γ-glytamyl-D-alanylamide and 2-oxo-3-sulfaminopropionic acid by hydrolysis with M hydrochloric acid [116,117]. The compound showed varying MIC from 0.4 µg/mL for *Escherichia coli* to 800 µg/mL for *Bacillus cereus*. Several other monobactams were isolated from *Chromobacterium violaceum* and other bacteria [117].

Aztreonam (Figure 21, **96**) is an example of a monbactam introduced on the market. The drug is derived from sulfazecin (**95**) by removal of the methoxy group and introduction of one of the side chain known from other β-lactam antibiotics [38]. It easily penetrates the outer membrane of Gram-negative aerobes such as *Pseudomonas aeruginosa*. Poor absorption from the gastro-intestinal duct makes intramuscular injection the preferred way of administration [38,118].

Carbapenemases particularly efficient in cleaving carbapenems have been discovered [77].

## 9. β-Lactamase Inhibitors

By far the major reason for resistance of bacteria against β-lactam antibiotics is expression of β-lactamases [72,77,119]. An important strategy for combating pathogenic resistant bacteria therefor is to inactivate β-lactamases by co-administration of β-lactam antibiotic drugs and β-lactamase inhibitors. Carbapenems are less sensitive to β-lactamases due to the different stereochemistry on the lactam ring [101]. The use of combination therapy is complicated by different absorption, metabolism and kinetic of the antibiotic and the β-lactamases inhibitor. In principle, these parameters are unique for every patient. In most cases, however, the combination drugs are distributed in fixed ratios [120]. The inhibitors might be of the three well-known types the β-lactam type, the diazabicyclooctane type and the boronic acid type. Clinical results for the use of some approved β-lactam-β-lactamase inhibitor combinations have been reviewed [119,120].

### 9.1. The β-Lactam Type of β-Lactamase Inhibitors

The first inhibitor to be discovered was clavulanic acid (Figure 22, **97**). It was discovered in a *Streptomyces* broth because growth of a penicillin-resistant *Klebsiella aerogenes* in a medium containing penicillin G (**24**) was inhibited. Thus, the presence of clavulanic acid (**97**) in the broth made *Klebsiella aerogenes* sensitive toward penicillin. [38,121,122]. The nucleus is an oxapenam isosteric to the carbapenam. The absence of a C-6 side chain deprives the molecule from a significant antibiotic effect but gives the molecule high affinity for β-lactamases [38]. Clavulanic acid inhibits serine β-lactamases but not metallo-β-lactamases [119]. Two combinations of amoxicillin (**44**) and clavulanic acid (**97**) in two different ratios (marketed as augmentin and timentin) are approved by the U.S. Food and Drug Administration (FDA) for treatment of infections with e.g., *Streptococcus pneumoniae*, *Haemophilus influenza, Moraxella catarrhailis, Klebsiella* spp., *S. aureus* and *Escherichia coli* [119]. Sulbactam (**98**) is a thiopenamdioxide with no C-6 side chain depriving an antibiotic effect [38]. It is approved to be used in combinations in different ratios with ampicillin (**42**) and cefoperazone for treatment of infections of *S. aureus, E. coli, Klebsiella* spp. and ***Enterobacter***
*cloaceae* [119]. In 1993 tazocin a combination of tazobactam (**99**) and piperacilllin (**49**) was introduced to treat infections with *S. aureus, S. pneumoniae* and *E. coli* [119]. Enmetazobactam (**100**) possesses a quaternary ammonium group, which together the carboxylate result in a net neutral compound with enhanced cell permeability. The enzyme group also increases affinity to spectrum β-lactamase of class A and D [119]. The β-lactamase synergistically augment the effects of penicillins sensitive to β-lactamases [58].

Sulbactam (**98**) is asumed to react irreversible with the β-lactamase as illustrated in Figure 12 [123]:

### 9.2. The Diazabicyclooctane Type of β-Lactamase Inhbitors

Design of the diazabicyclooctane group was inspired by the naturally occurring lactivicin (Figure 13, **101**) isolated from the broth of *Empedobacter lactamgenus* and *Lysobacter albus* [124,125]. The compound is a non-ribosomal dipeptide, in which the skeleton is derived from a serine and a glutamic acid. In solution an equilibrium between the two epimers at the quaternary carbon in the isoxazolidin-2-one ring is established [126]. Even though lactivicin is not a β-lactam but an isoxazilidin-2-one it stills react with penicillin-binding proteins and was found to be active against *Streptococcus pneumoniae* strains. The compound is highly active against Gram-positive bacteria and has modest activity against Gram-negative bacteria [125]. Other derivatives such as the phenoxy derivative **102** were found to be active against several clinically relevant Gram-negative strains such as *Pseudomonas aeruginosa* [58]. The discovery of lactivicins ability to acylate penicillin-binding peptides led to test of other heterocycles to find new heterocycles able to inactivate penicillin-binding peptides.

Soon it was observed that the imidazolin-2-one residue in the diazabicyclo[3.2.1}octane also reacts with β-lactamases. Consequently, synthesis and testing of similar compounds was initiated in several pharmaceutical companies. As result of these efforts avibactam (Figure 23, **103**) was discovered [58]. Avibactam has antibacterial effect but is approved for clinical use in combination with ceftazidime (**71**) (Avycaz) [58,127]. A diazbicyclooctane such as **104** has antibacterial activity [58].

The mechanism for the reaction of avibactam (**103**) with penicillin-binding peptides (PBP) is depicted in Figure 14. Like in the β-lactams a reactive amide, in this case a carbamate, is the functional group. The presence of a sulfate ester makes the nitrogen a better leaving group. The hydrophilic carbamoyl group makes a more stable complex with the serine-β-lactamases than do other β-lactam lactamase inhibitors such as clavulanic acid [58]. The carbamoyl-lactamase complex is surprisingly stable [58].

Ceftazidime (**71**) and avibactam (**103**) as combination therapy has been introduced to meet the urgent need for antipseudomonal antibiotics [128].

### 9.3. Boronic Acid β-Lactamase Inhibitors

The first boronic acid β-lactamase inhibitor to be used in the clinic was vaborbactam (Figure 24, **105**) in combination with meropenenem (**84**). The 1:1 combination is marketed as varbomere cures urinary tracts infections caused by Enterobacteriaceae resistant against the normally used drugs such as cephalosporins, fluoroquinolones and carbapenems [129]. Varbomere is also used against infections of the kidneys (pyelonephritis) [119]. Resistance is caused by expression of β-lactamases [129]. Boronic acid mimics the carbonyl group in the β-lactam ring. The mechanism of action involves a covalent binding of the boron atom with the active site serine hydroxyl group (Figure 25). Vaborbactam inhibits class A and class C serine β-lactamases [119]. The affinity for the enzyme can be adjusted by modifying the side chain on the boron containing ring [130].

Taniborbactam (**106**) is the first serine-β-lactamase inhibitor, which inhibits all the three groups of serine-β-lactamases, A, C and D and even some metallo-β-lactamase inhibitors, group B [119,120]. An apparently successful clinical trial 3 randomized double blind study comparing taniborlactam-cefepime versus meropenem is performed [120].

## 10. Conclusions and Perspectives

The observation that the major fall in the death rate caused by infectious diseases occurred before the introduction of antibiotic drugs with improved hygiene and sanitation [1] should not lead to the misconception that these drugs have not revolutionized the treatment of bacterial infections. The first antibiotic drugs, salvarsan and sulfonamides, had a limited spectrum of activity but still enabled physicians to cure previous untreatable diseases. Introduction of β-lactam antibiotics has significantly improved the armamentarium for treatment to such an extent that today less than 1 % of deaths are caused by of infectious diseases in high income countries [2,72,123]. However, development of extended spectrum resistant bacteria threatens the present state of control of infectious diseases [84]. In particular, a rapid spread of carbapenem resistant *Pseudomonas aruginosa* and extended spectrum resistant β-lactamase expressing species of Enterobacteriaceae is alarming [21]. In South Europe, 25–50% of *p. aeruginosa* strains are carbapenem resistant. Similar alarming observations have been made for *Acinetobacter baumanii* and *S. aureus* [131]. A meta-analysis has revealed that 14% of healthy individuals carry extended spectrum resistant β-lactamase producing bacteria. This number might increase [21]. In 2016 the annual mortality caused by resistant bacteria was estimated to 23,000 persons in the USA [70]. Treatment of patients infected with multi resistant bacteria is time consuming, expensive and risky. An analysis of 1391 patients infected with multi resistant bacteria hospitalized at Chicago Teaching Hospital disclosed (1) an excess duration of hospitalization estimated to between 6.4 to 12.7 days (2) additional costs of USD 18,600–29,000 for each patient (3) a mortality of 6.5% and (4) social costs of USD 11–15 millions [132]. An estimate predicts that in 2050 extended spectrum resistant bacteria cause 10 million deaths and will drain the society for USD 100 trillion each year [21].

A part of the problem origins in misuse of and overuse of antibiotics in human and veterinary medicine. Use of antibiotics to increase growth rate of livestock should be avoided. Recent estimates suggest that 70% of antibiotics are used in animals [131]. Public education should prevent use of antibiotics to treat infections caused by virus like common cold. Viral infections are not affected by antibiotics. Sale of antibiotics over the counter might have led to misuse and should be prohibited. Already Alexander Fleming in 1945 in his Nobel lecture warned against the risk of resistance by misuse of antibiotic drugs [131].

Despite spreading resistance only few new antibiotics are currently marketed, as illustrated in Table 1 [133].

Many multinational pharmaceutical companies have stopped their projects for developing antibiotics [70,133]. The vast majority of the drugs approved in the period 2014 to 2020 originated in well-known classes of antibiotics. Some innovative developments are seen in the field of β-lactamase inhibitors [84]. The poor return of investment compared to other drugs such as drugs for diabetes or cancer is demotivating. The sale of marketed antibiotic drugs the first two years afforded a sale of between USD 20 million (orbactiv, a glycopeptide for treatment of severe infection with Gram-positive bacteria) to USD 80 million (avycaz (Section 9.2)). In contrast, the sale of other medicines varied between USD 600 million (linzess/linacloride for treatment of constipation) and USD 1,500 million (januvia/sitagliptin for treatment of type 2 diabetes). The present spreading of resistance could result in a situation with increased mortality caused by communicative infections. To prevent such a situation the US Congress passed legislation in 2012 to create incentives for developing drugs [20,84]. Common for the antibiotic drugs approved in the period 2011 to 2017 is that they all have been marketed by small pharmaceutical companies; even though some of drugs were developed by international companies. Thus, small companies have a niche for developing and marketing antibiotic drugs. The latest marketed drugs are combination therapy combining β-lactams with β-lactamase inhibitors e.g., ceftazidime (**70**) in combination with avibactam (**103**) [120,134]. No compounds with a new mechanism of action have been marketed.

Beside antibiotics, vaccination is an additional tool to control communicative diseases. The first vaccination was performed in 1796 when Jenner injected lymph from a cow-pox vesicle into a boy. The boy developed a cow-pox pustule and became protected against smallpox [3]. Today vaccination has become an important tool for preventing diseases. It has led to the eradication of smallpox, the elimination of polio from most countries, and the control of other diseases, including diphtheria, tetanus, pertussis, rubella, and hepatitis B [135]. In addition to improving life quality it also saves the society for huge expenses to health care. An outstanding example is the present COVID-19 pandemic. Populations with high immunity because of vaccination carry a much smaller burden than poorly vaccinated populations. Since no person can be vaccinated against all infectious diseases antibiotic drugs always will be an important armamentarium in medicine.

In the fight against malaria, another severe burden for mankind [26], the assumption that man may defeat Nature turned out to be naive. This also is true for the fight against infectious diseases. Nature always fights back. That is why Nature has become so elaborate. The challenge is continuously to develop innovative drugs that keep the pressure from pathogenic organisms at a minimum, even though the return of investment might be poor. Governmental and international institutions and academia have to develop programs to control the spreading of extended spectrum resistance bacteria.

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
