# Peer review of "Drugs That Changed Society: History and Current Status of the Early Antibiotics: Salvarsan, Sulfonamides, and β-Lactams"

_molecules, 2021, doi:10.3390/molecules26196057_

Round 1

Reviewer 1 Report

I enjoyed reading the review by dr. Christensen which is well written and well organized. Examples have been chosen obviously with great attention and serve well the scope of sustaining the conclusion drawn by the Author. The historical perspective is not only interesting per se but again it severs to follow the development and impact of antibiotics discovery, therefore learning the "relative" impacts of different approaches and better planning future actions. It is also highly appreciated that the impact of drug/antibiotcs discovery has been correctly framed within a more general picture where also others interventions in public health were recognized as having had a huge impact in reducing mortality. 

I only have two (minor) points/suggestions:

  1. The manuscript can be improved in English and typos are present ("sees" instead of "seas", commas required, spaces between words missing, ect); minor intervention is however required;
  2. Since the Author, in the Perspective section, mentions about the double action of "vaccine development" and "drug/antibiotics development" and since this topic has caused in the Scientific Community some, unnecessary, competitions for funding, I think it could be useful to elaborate more (just in the same section) on this. Indeed, the two approaches (together with efficacious publish health policies) are not in competition and are rather complementary and should both be pursued and funded. This is particularly true for parasite-caused infection (e.g. malaria) or of course for viral infections but it is also applicable to bacteria where, for example, meningococcal vaccines proved efficient without for this reason, ruling out the relevance and effectiveness of antibiotics. On other hand, for example, the failure (so far) to develop a vaccine for tuberculosis whit the concomitant (partial) success in introducing new anti-TB drugs in the market does not rule out the relevance of vaccine research. I think that an elaboration of this concept, will add to this nice review.  

Author Response

I appreciate the reviewer’s comments on my manuscript.

Point 1: I have changed the language considerable to improve the language and to make the text easier to read. Please see the copy of the manuscript in which the corrections still are visible.

Point 2: I have added some sentences in Conclusions and Perspecctives: “Beside antibiotics, vaccination is an additional tool to control communicative diseases. The first vaccination was performed in 1796 when Jenner injected lymph from a cow-pox vesicle into a boy. The boy developed a cow-pox pustule and became protected against smallpox [3]. Today vaccination has become an important tool for preventing diseases. It has led to the eradication of smallpox, the elimination of polio from most countries, and the control of other diseases, including diphtheria, tetanus, pertussis, rubella, and hepatitis B [135]. In addition to improving life quality it also saves the society for huge expenses to health care. An outstanding example is the present Covid 19 pandemic. Populations with high immunity because of vaccination carry a much smaller burden than poorly vaccinated populations. Since no person can be vaccinated against all infectious diseases antibiotic drugs always will be an important armamentarium in medicine”.

Reviewer 2 Report

This review looks at the early antibiotics: salvarsan, sulfonamide and β-lactams, and their effects on society.  Whereas this is a good idea for a review article, I feel that it falls short in a number of respects and needs substantial re-writing before it can be acceptable for publication.  These issues are outlined below:

  1. Overall, the article is not well-written and contains numerous errors of English and typing that must be rectified. Ideally it should be proofed by a native English speaker.  There are too many examples of this to list them all, but here are the ones I spotted on pages 1-2:
    1. Line 5: tubed = piped
    2. Line 31: 1830th = 1830s. (This error appears many times.)
    3. Line 32: meant = suggested.
    4. Line 34: peal = peel.
    5. Line 44: ‘appropriate media to give organism’ – it is not clear what is meant here.
    6. Line 61: bathes = baths.
    7. Line 64: ‘After the antique…’ = After these times?
    8. Line 67: ..the sanitary idea?
    9. Line 69: sees = seas
    10. Line 85: are = is
    11. Line 93: and = an

There are many more errors of this type throughout the ms.

  1. The end of the Abstract is very abrupt and needs more details.
  2. Lines 21-24: it is hard to see what point the author is making here.
  3. The author needs to consider very carefully all the figures showing compound structures and reaction schemes, as to whether they are relevant to the ms.
  4. I feel that the author should consider carefully the content of the ms and ask whether it is addressing the title of the review. Personally I would prefer a title like: ‘History and current status of the early antibiotics salvarsan, sulphonamides and β-lactams, drugs that changed society’.
  5. I found the organisation and sub-heading of the review to be quite haphazard, and I would suggest the following:
  6. Introduction
  7. Salvarsan

2.1. Discovery and production

2.2. Mechanism of action

2.3, Resistance

2.4. Current status

(Repeat above sub-heads for sulphonamides and β-lactams)

  1. Conclusions and perspectives

Author Response

Point 1: I have changed the language considerable to improve the language and to make the text easier to read. Please see the copy of the manuscript in which the corrections still are visible.

Point 2: I have added some sentences to the Abstract: “The effects on life expectancy and life quality of these new drugs are indicated.”

Point 3: Please see Point 1.

Point 4: I have reconsidered all the figures.

Point 5: I have changed the title to “Drugs that changed society: History and current status of the early antibiotics, salvarsan, sulfonamides and β-lactams”

Point 6: I have changed the heading and subheadings as suggested.

Round 2

Reviewer 2 Report

This is a revised version of a review that looks at the early antibiotics: salvarsan, sulfonamide and β-lactams, and their effects on society.  This is a somewhat idiosyncratic article that nonetheless has merit and will be a value to those working in the field.  Since my original review the author has made some changes but there are still problems that need resolving.  The main issue the English, and the ms needs extensive editing before it can be published.  The author has corrected some errors but there are still a great deal more that need addressing, throughout the ms..  These issues are outlined below:

  1. Again the ms needs correcting in terms of English and I again list errors spotted only on pages 1-2; there are many more elsewhere in the ms:
    1. Line 9: treatment of diseases
    2. Line 21: in millennia = for millennia
    3. Line 23: over morbidity = of morbidity
    4. Line 27: standard = standards
    5. Line 36: meant = suggested
    6. Line 43: origin = originate
    7. Line 50: 1882
    8. Line 55: Mycobacterium tuberculosis
    9. Line 70: extend = extent
  2. Other changes seem to have been done. The author did not adopt the suggested changes in sub-headings but that is a matter of personal choice.

Author Response

An extensive revision of the language has been perfomed.